# Enhancing Object Discovery for Unsupervised Instance Segmentation and Object Detection

## Abstract

We propose **C**ut-**O**nce-and-**LE**a**R**n (COLER), a simple approach for unsupervised instance segmentation and object detection. COLER first uses our developed CutOnce to generate coarse pseudo labels, then enables the detector to learn from these masks. CutOnce applies Normalized Cut only once and does not rely on any clustering methods, but it can generate multiple object masks in an image. We have designed several novel yet simple modules that not only allow CutOnce to fully leverage the object discovery capabilities of self-supervised models, but also free it from reliance on mask post-processing. During training, COLER achieves strong performance without requiring specially designed loss functions for pseudo labels, and its performance is further improved through self-training. COLER is a zero-shot unsupervised model that outperforms previous state-of-the-art methods on multiple benchmarks. We believe our method can help advance the field of unsupervised object localization. **Code is in the supplementary materials.**

## 1 Introduction

In computer vision, segmentation tasks heavily rely on large-scale manual annotations, which significantly limits the development speed of the field. To reduce dependence on labeled data, researchers have begun exploring weakly-supervised or unsupervised approaches (Simeoni et al., 2025). This work focuses on exploring how to perform unsupervised instance segmentation and object detection efficiently. Found (Siméoni et al., 2022) established the two-stage framework for unsupervised segmentation: generating pseudo labels followed by training a detector using them.

In recent years, methods that use self-supervised models (SSM) to extract image features and then apply Normalized Cut (NCut) (Shi & Malik, 2000) to generate pseudo labels have made impressive progress, outperforming many other approaches. TokenCut (Wang et al., 2022b), MaskCut (Wang et al., 2023), and VoteCut (Arica et al., 2024) all use the DINO (Caron et al., 2021) model to extract features and produce reliable pseudo labels, achieving significant advances in unsupervised object localization field. DiffCut (Couairon et al., 2024) employs a diffusion UNet (Ronneberger et al., 2015) encoder for feature extraction, targeting unsupervised semantic segmentation.

The NCut paper (Shi & Malik, 2000) recommends two approaches for extending from single-object to multi-object segmentation. One approach is to perform NCut once, using the top-$n$ eigenvectors as an $n$-dimensional indicator vector, followed by clustering (e.g., K-Means) to segment multiple objects. VoteCut (Arica et al., 2024) adopts a clustering-based approach and generates masks using only the second smallest eigenvector. However, clustering methods require specifying the number of clusters, which reduces the generality of such approaches. The other approach is to recursively partition the separated groups, that is, to further split the current foreground or background regions. MaskCut (Wang et al., 2023) adopts this strategy by recursively partitioning the background. However, this approach clearly suffers from error accumulation as the number of recursive steps increases. These two types of methods have low computational efficiency, hindering their application and further exploration in related fields.

This paper uses NCut to segment multiple objects, but it differs from the two types of methods mentioned above. *Our CutOnce neither relies on multiple applications of NCut nor on clustering methods when discovering multiple objects.* In short, by applying NCut only once and designing two object discovery enhancement modules, CutOnce can segment multiple objects from an image,

which is the origin of its name. Additionally, with a rank feature filter, *the method can detect over 10 object masks (examples are shown in the Appendix).* CutOnce does not require computationally expensive methods such as Conditional Random Field (CRF) (Krähenbühl & Koltun, 2011). By fully leveraging the feature, geometric, and localization cues learned by SSM, it achieves promising results. Table 1 summarizes the key properties of CutOnce and popular existing methods (Wang et al., 2023; Arica et al., 2024), showing that CutOnce not only detects more objects

Table 1: Key properties of our CutOnce and COLER with state-of-the-art methods.

| Train-Free Mask Generators | MaskCut | VoteCut | CutOnce |
|---|---|---|---|
| Normalized Cut #nums | 3 | 1 | 1 |
| clustering method | × | ✓ | × |
| post-process mask | ✓ | ✓ | × |
| self-supervised #models | 1 | 6 | 1 |
| max #objects detected | 3 | 10 | >**10** |
| mask generation time (s/img) | 5.6 | 2.4 | **0.24** |

| Pseudo Mask Learners | CutLER | CuVLER | COLER |
|---|---|---|---|
| pseudo mask loss function | ✓ | ✓ | × |
| $AP_{50}^{mask}$ on COCO val2017 | 18.9 | 19.3 | **20.1** |

but also generates annotations up to $10\times$ faster. COLER uses CutOnce's annotations for training and achieves good performance through self-training alone, without the need to design specialized loss functions to handle errors in the "ground truth" provided by pseudo masks.

The contributions of this paper can be summarized as follows: *1)* We develop an efficient tool, CutOnce, for generating coarse masks. *2)* We train a detector, COLER, using pseudo masks generated by CutOnce. COLER is a zero-shot model that outperforms previous state-of-the-art(SOTA) methods across multiple datasets.

## 2 RELATED WORK

**Self-Supervised Vision Transformer.** Self-supervised models are capable of learning deep features without human annotations or supervision. ViT (Dosovitskiy et al., 2020) captures long-range dependencies between different regions in images through a global self-attention mechanism, making it easier to focus on semantically consistent target regions compared to CNN. DINO (Caron et al., 2021) combines both advantages, playing a key role in advancing unsupervised object localization. It adopts a teacher-student training framework and introduces a novel contrastive learning strategy that compares features from the original image and its random crops to learn stronger visual representations. Thanks to the built-in spatial attention mechanism of the ViT architecture, DINO's attention maps can be directly used for localization and have shown advantages over previous methods. By further processing DINO's attention maps, more precise object regions can be obtained. And many methods (Wang et al., 2022b; 2023; Arica et al., 2024; Sick et al., 2024) have been developed based on this idea, achieving significant progress in their respective fields.

**Unsupervised Instance Segmentation and Object Detection.** The following methods all utilize unsupervised object localization, which is then applied to instance segmentation and object detection tasks. Early approaches do not use NCut (Shi & Malik, 2000), while later ones adopt it.

LOST (Siméoni et al., 2021) is the first to localize objects by leveraging the final layer CLS token from a pre-trained transformer DINO (Caron et al., 2021) and computing patch-wise similarity within single image, but it can only localize one object. MOST (Rambhatla et al., 2023) extends this to multiple objects through entropy-based box analysis and clustering. FreeSOLO (Wang et al., 2022a) uses features from DenseCL (Wang et al., 2020) to generate a set of "queries" and "keys" which are convolved to produce masks and also supports multiple object localization. These methods do not rely on NCut and their robustness on large-scale datasets remains unconvincing.

TokenCut (Wang et al., 2022b) is the first to apply NCut to features extracted by DINO, significantly improving the quality of pseudo labels, but it can only segment single instance. CutLER (Wang et al., 2023) recursively applies NCut on one image to generate masks for multiple instances. CuVLER (Arica et al., 2024) uses multiple self-supervised models to generate diverse mask proposals and selects the best masks through clustering and pixel-wise voting. In addition, it assigns a confidence score to each pseudo mask. Table 1 presents the properties of the above methods and compares them with ours. CutS3D (Sick et al., 2024) introduces 3D information to enhance segmentation in 2D images, showing certain advantages in handling overlapping or connected objects and

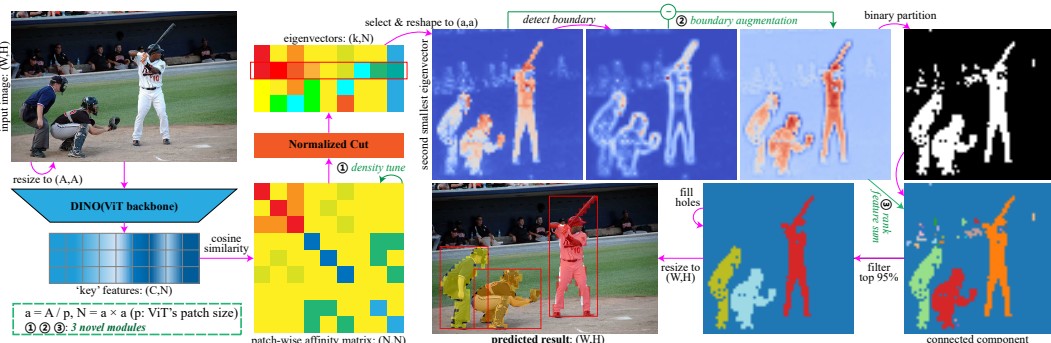

Figure 1: **Overview of CutOnce.** First, the resized image is processed by DINO to extract the "key" features. Then, Construct the affinity matrix and apply the NCut algorithm to obtain the second smallest eigenvector. Next, the original eigenvector is used to compute the boundary eigenvector, and the two are subtracted element-wise to produce the boundary-enhanced eigenvector. Finally, perform graph partitioning on the enhanced eigenvectors to generate segmentation masks.

demonstrating strong potential for real-world applications. DiffNCut (Liu & Gould, 2024) proposes Differentiable Normalized Cuts. In other words, it uses NCut to propagate gradients and fine-tune DINO. Since NCut enhances object discovery, methods that use it outperform those that do not.

## 3 METHOD

This chapter introduces a novel pipeline called **Cut-Once-and-LEaRn** (COLER), designed for unsupervised instance segmentation and object detection. We first propose CutOnce, a training-free method that efficiently generates masks for multiple objects. Built upon prior work, CutOnce enhances object discovery by optimizing the edge weighting scheme and introducing a boundary augmentation strategy in the NCut algorithm. In addition, it incorporates a rank feature filter to retain a sufficient number of valuable object masks. Finally, we train a detector using the masks generated by CutOnce and further improve its performance through self-training.

### 3.1 PRELIMINARIES

**Normalized Cut.** NCut (Shi & Malik, 2000) formulates image segmentation as a graph partitioning problem. It constructs a fully connected undirected graph $\mathbf{G} = (\mathbf{V}, \mathbf{E})$ by representing the image as a set of nodes, where each pair of nodes is connected by an edge with weight $w_{ij}$ indicating their similarity. NCut minimizes the cost of partitioning the graph into two subgraphs by solving a generalized eigenvalue system

$$(\mathbf{D} - \mathbf{W})\mathbf{x} = \lambda \mathbf{D}\mathbf{x} \tag{1}$$

to yield a set of $N \times N$ eigenvectors $\mathbf{x}$, where $N$ denotes the number of nodes. Here, $\mathbf{D}$ is an $N \times N$ diagonal matrix with $d_{ii} = \sum_j w_{ij}$, and $\mathbf{W}$ is an $N \times N$ symmetric matrix representing the adjacency matrix of edge weights.

**TokenCut and MaskCut.** Our CutOnce is *based on the overall workflow of TokenCut* (Wang et al., 2022b), with some implementation details *adopting the design of MaskCut* (Wang et al., 2023). The following describes the consistent parts of CutOnce with existing methods.

First, the input image is passed through a single self-supervised model to extract the "key" features from the last attention layer, denoted as $\mathbf{K} \in \mathbb{R}^{D \times N}$, where $D$ is the feature dimension and $N$ is the number of nodes. The key feature of each patch is represented as a feature vector $\mathbf{k}_i$ ($i = 1, \ldots, N$). These features encode the spatial information captured by ViT (Dosovitskiy et al., 2020), so the cosine similarity between them can be used to calculate the elements in $\mathbf{W}$.

$$w_{ij} = \cos(\mathbf{k}_i, \mathbf{k}_j) = \frac{\mathbf{k}_i^T \mathbf{k}_j}{\|\mathbf{k}_i\|_2 \|\mathbf{k}_j\|_2} \tag{2}$$

Then, the second smallest eigenvector $\mathbf{y}_1$ is obtained from the solution of Equation 1, which can be viewed as *an enhanced attention map*. When the splitting threshold is set to $\overline{\mathbf{y}_1} = \frac{1}{N} \sum_i \mathbf{y}_1^i$, $\mathbf{y}_1$ can be effectively divided into background and foreground. To determine which group corresponds to the foreground, we examine the distribution of $\mathbf{y}_1$ on the ImageNet Deng et al. (2009) and COCO Lin et al. (2014) datasets, using criteria similar to MaskCut: *1)* The foreground set contains fewer than three of the four image corners, an idea inspired by the object-centric prior Maji et al. (2011). *2)* $|\max(\mathbf{y}_1)| > |\min(\mathbf{y}_1)|$. If either of these conditions is not satisfied (with condition 1 taking priority), the feature vector $\mathbf{v}$ used for mask partitioning is set to $-\mathbf{y}_1$; otherwise, it is set to $\mathbf{y}_1$. Finally, the groups in $\mathbf{v}$ with values greater than the partition point are regarded as foreground, while those with smaller values are regarded as background.

Existing methods have paved the way for our research, but the *following limitations* remain urgent to address: *1)* The NCut stage does not produce refined masks. *2)* Processing multiple targets is relatively time-consuming. *3)* Predicted masks are prone to errors.

## 3.2 CUTONCE: EFFICIENT MASK GENERATOR

Given the limitations of prior works, we aim to address the following challenges: *1)* How to produce more accurate foreground boundaries. *2)* How to enable NCut algorithm to discover multiple objects rather than focusing on a single one. *3)* How to segment more objects without introducing too many incorrect masks. Thus, we introduce three simple yet effective modules to tackle these issues.

**Three Novel Modules in CutOnce.** The overall pipeline of CutOnce is illustrated in Figure 1. The first two improvements optimize the *input* and *output* of the NCut algorithm, respectively. Their underlying principle is consistent: *enhancing the distinction between foreground and background to make the eigenvector distribution closer to the ideal*, resembling the concept of contrastive learning. The third improvement targets the *graph partition* stage, effectively filtering out the most salient multiple objects.

**Density-Tune Cosine Similarity.** Previous methods (Wang et al., 2022b; 2023; Arica et al., 2024) compute the edge weight matrix $\mathbf{W}$ solely based on the cosine similarity between nodes, ignoring the variations in feature density across different image regions. This often leads to over-activation in certain areas, which negatively affects boundary localization. To address this issue, we propose a local-density-aware temperature modulation for cosine similarity. The idea is to adaptively adjust the temperature parameter in similarity computation based on the local density of feature points.

For the convenience of subsequent calculations, the deep learning features $\mathbf{K}$ are first normalized. The elements of the adaptive edge weight matrix are defined as:

$$w_{ij} = \frac{\cos(\mathbf{k}_i, \mathbf{k}_j)}{T_{ij}} = \frac{\mathbf{k}_i^T \mathbf{k}_j}{T_0 + \alpha \cdot \frac{\rho_i + \rho_j}{2}} \tag{3}$$

where $T_{ij}$ denotes the adaptive temperature parameter, $T_0$ is the base temperature, $\alpha$ is the modulation parameter, and $\rho_i$ and $\rho_j$ represent the local densities of feature points $i$ and $j$, respectively. The local density is computed by first calculating the pairwise cosine similarity matrix $\mathbf{S} = \mathbf{K}\mathbf{K}^T$ for all patches in batch. Then, for each feature point $i$, we select its top-$k$ most similar neighbors (excluding itself) and compute the local density as:

$$\rho_i = \frac{1}{k} \sum_{j \in \mathcal{N}_k(i)} \mathbf{S}_{ij} \tag{4}$$

where $\mathcal{N}k(i)$ denotes the set of indices corresponding to the $k$ most similar samples to the $i$-th sample (excluding $i$ itself). The modulated $\mathbf{W}$ still requires feature contrast enhancement, following the approach in TokenCut (Wang et al., 2022b). Specifically, $\mathbf{W}_{ij}$ is set to 1 if $\mathbf{W}_{ij} \geq \tau^{\text{ncut}}$, and to $1e^{-5}$ otherwise, where $\tau^{\text{ncut}}$ is set to 0.15 by default.

Background regions typically exhibit sparse and relatively uniform features, allowing low-temperature areas to preserve the original similarity. In contrast, object interiors tend to have dense but uneven features, where high-temperature areas suppress similarity more strongly, making intra-object similarities more consistent. Obviously, regions with uniform similarity are more likely to be grouped together, and our improvement leads to more accurate separation of foreground and background.

A similar idea is adopted by the clustering algorithm LDP-SC (Long et al., 2022), which combines local density peaks with NCut and demonstrates significant advantages when handling locally tree-structured data. This further validates the effectiveness and rationality of integrating density information into graph cut methods.

**Boundary Augmentation.** The attention map $\mathbf{y_1}$ output by NCut tends to focus on a single object, which is the fundamental reason why MaskCut (Wang et al., 2023) applies the NCut algorithm multiple times to discover multiple objects. However, our goal is to obtain masks for multiple objects using NCut only once. This raises an important question: Are the less salient objects in the foreground being ignored by the SSM or the NCut algorithm? In fact, the potential objects are already represented, but it is difficult to assign those with relatively low attention to the foreground. From the first attention map (visualization of $\mathbf{y_1}$) in Figure 1, the following information can be easily extracted: *1)* Regions with larger feature values are usually concentrated within parts of the objects. *2)* In some object boundary areas, the feature values differ significantly from their surrounding regions.

To segment more objects, the salient regions within the foreground need to be expanded. Can boundary information be leveraged to achieve this? In practice, incorporating boundary information to refine original eigenvector $\mathbf{y_1}$ has proven to be an effective approach. We propose a boundary-enhanced feature representation:

$$X_a = X - X_b \tag{5}$$

where $X$ is the original eigenvector, and $X_b$ is the boundary eigenvector obtained by calculating the difference between each point and its neighborhood:

$$X_b = \frac{1}{k} \sum_{n \in \mathcal{N}_k} |X - X_n|, \quad k \in \{4, 8\} \tag{6}$$

Here, $\mathcal{N}_k$ denotes the $k$-neighborhood, which is set to 8 by default, and $X_n$ represents the feature values within the neighborhood. To correctly compute boundary pixels, padding is applied to the four edges and four corners of the feature map. The padding regions should not introduce new groups, so the original boundary features are simply extended. Specifically, the feature values in the padding areas are set to those of the adjacent boundary pixels.

The third attention map in Figure 1 demonstrates the benefits of this improvement: *(1) The saliency of more objects is enhanced. (3) Nearby objects are less likely to be considered as a single entity.*

First, we analyze the first advantage. $\mathbf{X}_b$ places high attention on regions with large feature differences, particularly on small areas around the boundary between foreground and background, as well as certain regions inside the foreground. After optimizing the attention distribution using Equation 5, $\mathbf{X}_a$ exhibits *generally reduced attention within the foreground* compared with $\mathbf{X}$, leading to smaller feature differences within the foreground. As a result, *the salient regions of the foreground become larger*. Next, we analyze the second advantage. $\mathbf{X}_b$ shows high attention on both sides of object boundaries, which causes $\mathbf{X}_a$ to "merge" the areas around the boundary into the background, *making all detected objects smaller*. For objects that are close to each other, the gaps between them are enlarged, which facilitates their correct separation. However, for small objects, prediction errors increase. If an object is very small in area, this may cause such "noisy points" to vanish. Considering both comprehensive theoretical analysis and subsequent ablation study, boundary enhancement proves to be a strategy with more advantages than disadvantages. *More visualizations of the computation process for this module are provided in the Appendix.*

**Ranking-Based Instance Filter.** After extracting the foreground region from eigenvector using a segmentation threshold (as described in the *Preliminaries* section), the next step is to separate multiple objects from the foreground. To achieve this, we first apply 4-connectivity to perform connected component decomposition on the foreground region, treating each connected component as a candidate object. To select multiple salient objects from these candidates, we propose a feature rank-based connected component filtering strategy, which proceeds as follows:

1. Sort: Suppose there are $N$ candidate object regions. Let $s_i$ denote the feature sum of the $i$-th region. Sort all feature sums $\{s_i\}_{i=1}^N$ in descending order to obtain the index sequence $\{i_1, i_2, \ldots, i_N\}$.

2. Cumulative screening: Select the top-ranked objects one by one until the cumulative feature proportion reaches a predefined threshold: $\frac{\sum_{j=1}^{k} s_{i_j}}{\sum_{i=1}^{N} s_i} \geq \tau$, where $\tau \in (0, 1)$ is the feature preservation threshold.

3. Output: The masks corresponding to the top $k$ selected objects.

Obviously, this strategy can extract the salient objects in the foreground in descending order of their feature significance. More importantly, it introduces only a single hyperparameter.

### 3.3 SELF-TRAINING.

After training the detector with pseudo labels generated by CutOnce, the detector can identify more masks than those in the pseudo labels. Therefore, we adopt a self-training strategy to further improve the model performance. In the $t$-th round of self-training ($t \in 1, 2, \ldots$), we first perform inference on the training data using the current model, retaining predicted masks with confidence scores higher than $0.6 - 0.05t$ as high-quality pseudo-labels. To avoid label duplication and maintain diversity in the training data, we also select a subset of pseudo-labels from round $(t-1)$ whose IoU with the current high-confidence predictions is $< 0.5$. The final training labels for round $t$ are obtained by merging these two sets.

## 4 EXPERIMENTS

This paper mainly uses $AP_{50}$ and AP as the evaluation metrics for presenting results. *Our method gets rid of the dependence on CRF (Krähenbühl & Koltun, 2011), but using CRF to post-process pseudo-labels can further improve the results.* When CutOnce uses CRF, it is called CutOnce*, and when COLER uses CutOnce*, it is called COLER*.

### 4.1 IMPLEMENTATION DETAILS

**Datasets.** In this paper, only the images from the validation set of ImageNet-1K (Deng et al., 2009) (50K images) are used for all training processes of the COLER model, with no manual annotations or any supervised pre-trained models employed in the training.

We evaluate on two subsets of the COCO (Lin et al., 2014) dataset, LVIS (Gupta et al., 2019), VOC (Everingham et al., 2010), KITTI (Geiger et al., 2012), OpenImages (Kuznetsova et al., 2020) and Objects365 (Shao et al., 2019), resulting in a total of 7 benchmarks.

**CutOnce.** We resize images to 480×480 pixels and use the ViT-B/8 (Dosovitskiy et al., 2020) DINO (Caron et al., 2021) model by default to extract features. For the density-tune similarity module, $k$, $T_0$, and $\alpha$ are set to 10, 1.0, and 0.5, respectively. The filter parameter $\tau$ is set to 0.95

**CAD (class-agnostic detector).** All training and inference are conducted on a single NVIDIA RTX 4090 GPU. All experiments are implemented on the detectron2 (Wu et al., 2019) platform using Cascade Mask R-CNN (Cai & Vasconcelos, 2017) as the default detector. The detector is trained with masks and bounding boxes generated by CutOnce for 80K iterations with copy-paste augmentation (Ghiasi et al., 2021). Similar to the copy-paste augmentation in CutLER (Wang et al., 2023), we randomly downsample the masks using a scalar uniformly sampled between 0.3 and 1.0. The batch size is set to 8, learning rate to 0.01, weight decay to $5 \times 10^{-5}$, and momentum to 0.9.

**Self-Training.** In this stage, the model is initialized with the weights from the previous phase and trained for 60K iterations. Other settings follow CutLER, with the learning rate set to 0.005 and the copy-paste augmentation scalar uniformly sampled between 0.5 and 1.0.

**SOTA Comparison.** We compare our method with CutLER (Wang et al., 2023) and CuVLER (Arica et al., 2024), both of which satisfy the following criteria: *1)* publicly available code with reproducible results; *2)* support for object-level zero-shot instance segmentation.

### 4.2 PSEUDO LABELS EVALUATION

To evaluate the pseudo masks using the official COCO (Lin et al., 2014) evaluation tool, each annotation must be assigned a confidence score. *The score has a significant impact on AP but does*

Table 2: **Evaluation of pseudo labels.** #N denotes the average number of masks per image.

| Datasets → Methods | Use CRF | ImageNet val | | | | | COCO val2017 | | | | | | | | |
|---|---|---|---|---|---|---|---|---|---|---|---|---|---|---|---|
| | | $AP^{box}$ | $AP^{box}_{50}$ | $AP^{box}_{75}$ | $AR^{box}_{100}$ | #N | $AP^{box}$ | $AP^{box}_{50}$ | $AP^{box}_{75}$ | $AR^{box}_{100}$ | $AP^{mask}$ | $AP^{mask}_{50}$ | $AP^{mask}_{75}$ | $AR^{mask}_{100}$ | #N |
| MaskCut | ✓ | 10.6 | 20.3 | 10.0 | 27.7 | 1.9 | 3.9 | 7.9 | 3.3 | 7.7 | 3.1 | 6.8 | 2.5 | 6.5 | 1.9 |
| VoteCut | ✓ | **20.9** | **36.2** | **20.0** | **45.0** | 8.9 | **5.5** | **10.7** | **4.9** | **12.2** | **4.5** | **9.3** | **3.9** | **10.3** | 8.6 |
| CutOnce(ours) | ✗ | 16.5 | 32.5 | 15.0 | 31.5 | 1.8 | 4.1 | 8.2 | 3.6 | 7.6 | 3.1 | 7.0 | 2.4 | 6.0 | 1.8 |
| CutOnce*(ours) | ✓ | 16.9 | 32.6 | 15.4 | 32.3 | 1.8 | 4.2 | 8.2 | 3.7 | 7.9 | 3.4 | 7.2 | 2.9 | 6.8 | 1.8 |

*not affect AR.* Since VoteCut (Arica et al., 2024) comes with its own scoring mechanism, we retain its original setting. To ensure a relatively fair comparison, we apply the same scoring scheme to both MaskCut (Wang et al., 2023) and CutOnce. The reason is that both methods output masks in descending order of object saliency and treat all outputs as "ground truth". For multiple masks associated with the same image, we adopt a linearly decreasing score assignment scheme. The confidence score for each mask is defined as 1.0 if $N = 1$, and as $1.0 - \frac{k}{2N-2}$ if $N > 1$, where $N$ is the total number of masks in the image and $k$ is the index of the current mask ($k = 0, 1, \ldots, N - 1$). This ensures that the first mask always receives the highest confidence score of 1.0, while the last mask is assigned a score of 0.5.

Table 2 presents the quantitative comparisons of different methods, and Figure 2 shows the corresponding qualitative results. On ImageNet val, all metrics of CutOnce lie between those of MaskCut and VoteCut. On COCO val2017, CutOnce shows poor performance in $AR^{mask}_{100}$. From the performance of CutOnce*, it clearly outperforms CutOnce and MaskCut, achieving substantial improvements in all metrics except $AP_{50}$. Evidently, using CRF to refine the masks has a positive effect on high-precision localization, while having little impact on coarse localization, ultimately leading to significant gains

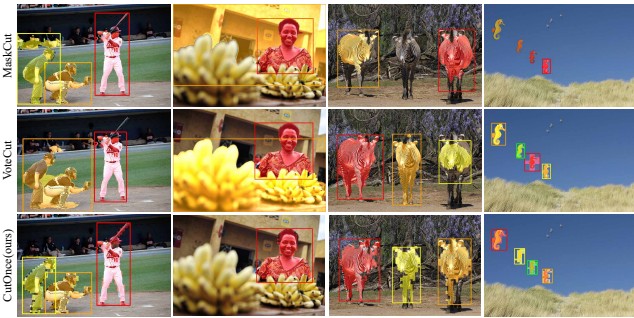

Figure 2: **Qualitative comparison between our CutOnce and related methods on COCO val2017.** VoteCut only displays predicted masks with confidence scores $\geq 0.5$, whereas other methods display all predicted masks.

in AP and AR. As shown in Figure 2, the targets localized by CutOnce are always correct. Since CutOnce does not use CRF for post-processing, its boundaries are slightly coarse. However, as illustrated in the second image, CutOnce identifies fewer targets than previous methods, a claim also supported by the average number of masks reported in Table 2. CutOnce applies NCut only once, *whereas NCut inherently tends to focus on a single target, even when we attempt to weaken this property.* MaskCut, which applies NCut three times, can localize up to three targets, while VoteCut integrates multiple self-supervised models and combines the different targets emphasized by these models. Overall, the pseudo labels generated by CutOnce contain relatively fewer noisy labels, which is beneficial for model learning.

Table 3: **Zero-shot evaluation across three COCO-based datasets.** IN and '1 + 3' denote ImageNet-1K and one training plus three rounds of in-domain self-training, respectively.

| Datasets → Methods | Pretrain | Train #Rounds | COCO 20K | | | | COCO val2017 | | | | LVIS | | | |
|---|---|---|---|---|---|---|---|---|---|---|---|---|---|---|
| | | | $AP^{box}$ | $AP^{box}_{50}$ | $AP^{mask}$ | $AP^{mask}_{50}$ | $AP^{box}$ | $AP^{box}_{50}$ | $AP^{mask}$ | $AP^{mask}_{50}$ | $AP^{box}$ | $AP^{box}_{50}$ | $AP^{mask}$ | $AP^{mask}_{50}$ |
| CutLER | IN train | 1 + 3 | 12.5 | 22.4 | 10.0 | 19.6 | 12.3 | 21.9 | 9.7 | 18.9 | 4.5 | 8.4 | 3.5 | 6.7 |
| CuVLER | IN val | 1 | 12.7 | 23.5 | 10.0 | 20.1 | 12.6 | 23.0 | 9.8 | 19.3 | 4.5 | 8.6 | 3.6 | 6.9 |
| COLER(ours) | IN val | 1 + 1 | 12.6 | 24.1 | 9.8 | **20.5** | 12.5 | **23.8** | 9.6 | 20.1 | 4.6 | 9.2 | 3.7 | 7.3 |
| COLER*(ours) | IN val | 1 + 1 | **12.9** | **24.2** | **10.2** | **20.5** | **12.8** | **23.8** | **10.0** | **20.2** | **4.9** | **9.4** | **3.9** | **7.4** |

Table 4: **Zero-shot unsupervised object detection evaluation.** Avg. denotes the average value.

| Datasets → Metrics → | Avg. AP$_{50}$ AP | COCO AP$_{50}$ AP | COCO20K AP$_{50}$ AP | LVIS AP$_{50}$ AP | VOC AP$_{50}$ AP | KITTI AP$_{50}$ AP | OpenImages AP$_{50}$ AP | Objects365 AP$_{50}$ AP |
|---|---|---|---|---|---|---|---|---|
| CutLER | 21.0 11.3 | 21.9 12.3 | 22.4 12.5 | 8.4 4.5 | 36.9 20.2 | 18.4 8.5 | 17.3 9.7 | 21.6 11.4 |
| CuVLER | 21.2 11.4 | 23.0 12.6 | 23.5 12.7 | 8.6 4.5 | **39.4 22.3** | 13.0 5.1 | **19.6 11.6** | 21.6 10.9 |
| COLER(ours) | 22.3 11.4 | **23.8** 12.5 | 24.1 12.6 | 9.2 4.6 | 39.1 20.5 | 20.8 8.8 | 16.7 9.3 | 22.6 11.2 |
| COLER*(ours) | **22.6 11.8** | **23.8 12.8** | **24.2 12.9** | **9.4 4.9** | 39.3 21.0 | **21.1 9.3** | 17.8 10.1 | **22.8 11.6** |

## 4.3 UNSUPERVISED ZERO-SHOT EVALUATIONS

We evaluate COLER on *7 different benchmarks*, containing a variety of object categories and image styles, to validate its effectiveness as a general unsupervised method.

**Detailed in COCO Datasets.** COCO 20K, COCO val2017, and LVIS are all from COCO and provide segmentation annotations, with the corresponding results reported in Table 3. On the other two COCO datasets, COLER achieves slightly lower AP than the previous best results, but exhibits the opposite trend in AP$_{50}$. Obviously, compared with other methods (Wang et al., 2023; Arica et al., 2024), COLER demonstrates strong competitiveness on the densely annotated LVIS dataset. *When CRF is incorporated into the COLER pipeline (COLER\*), the AP achieves the best performance on most datasets.* Figure 3 shows the qualitative results of COLER compared with related methods. Obviously, *COLER often detects more useful instances, including some that are not annotated in the ground truth.*

**Object Detection.** In Table 4, we report COLER's object detection performance across all datasets.

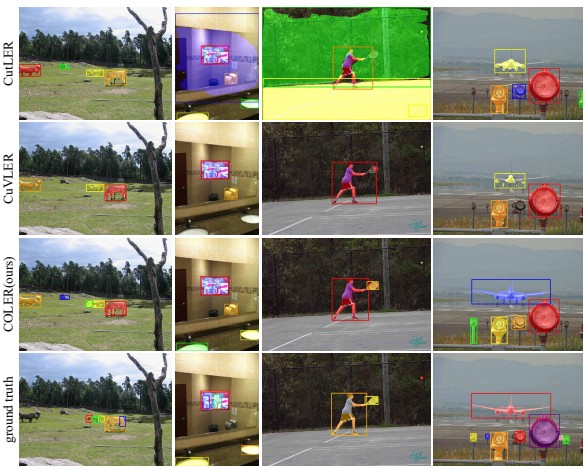

Figure 3: **Qualitative comparison between our COLER and SOTA methods on COCO val2017.** Only predictions with confidence $\geq 0.5$ are shown.

On average, COLER achieves a significant improvement in AP$_{50}$, while its AP remains nearly the same as previous methods. Comparing the best results on each dataset, COLER shows significant advantages on KITTI and LVIS, while performing the worst on OpenImages. COLER* further improves the prediction performance, which is consistent with the experimental results presented earlier. Overall, our method achieves certain advantages across all datasets.

| $\tau$ | 0.8 | 0.9 | **0.95** | 0.99 | $k$ | 3 | 5 | **10** | 20 | $T_0$ | 0.8 | **1.0** | 1.2 | $\alpha$ | 0.3 | **0.5** | 0.7 |
|---|---|---|---|---|---|---|---|---|---|---|---|---|---|---|---|---|---|
| AP$_{50}^{mask}$ | 17.3 | 18.5 | **19.2** | 17.1 | AP$_{50}^{mask}$ | 18.5 | 18.9 | **19.2** | 18.3 | AP$_{50}^{mask}$ | 18.1 | **19.2** | 18.5 | AP$_{50}^{mask}$ | 18.3 | **19.2** | 18.6 |

Table 5: **Ablation study of CutOnce hyperparameters on COCO val2017 .** $\tau$ denotes the preservation ratio in the filter, while $k$, $T_0$, and $\alpha$ are parameters related to the adaptive edge weight matrix.

## 4.4 ABLATION STUDY

Table 5 reports the ablation study of the hyperparameters introduced by CutOnce, where the detector is trained only once. Figure 4 visualizes the ablation study of two NCut refinement modules introduced in CutOnce for enhanced object discovery. It is evident that when the two modules are combined, CutOnce can finely segment the three most salient targets, whereas removing either module often results in poor mask boundaries for certain objects.

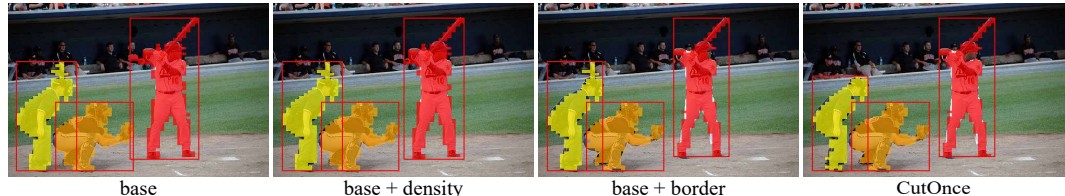

| base | base + density | base + border | CutOnce |

Figure 4: **Ablation study on the two NCut refinement modules in CutOnce.**

Table 6 reports the impact of each component in COLER on the final results across two datasets. Each component contributes to performance improvements to varying degrees, but boundary augmentation module proves to be the most critical for boosting performance. Additionally, copy-paste augmentation (Ghiasi et al., 2021) and self-training during the training process also contribute significantly to the overall improvement of COLER.

Table 7 reports the impact of the number of self-training rounds on the final results of COLER. The first round of self-training improves all metrics, and the second round shows no significant gains compared with the first. *COLER uses one round of self-training.*

Table 6: **Ablation analysis of COLER components on COCO val2017 and KITTI.**

| Methods | COCO | | KITTI | |
|---|---|---|---|---|
| | $AP^{mask}$ | $AP^{mask}_{50}$ | $AP^{box}$ | $AP^{box}_{50}$ |
| TokenCut + CAD | 5.7 | 13.5 | 5.2 | 13.6 |
| + rank feature filter | 7.5 | 15.7 | 6.3 | 15.2 |
| + similarity tune | 7.7 | 16.1 | 6.6 | 15.8 |
| + boundary augment | 8.6 | 18.0 | 7.5 | 18.6 |
| + copy-paste | 9.2 | 19.6 | 7.6 | 18.8 |
| + self-training (**COLER**) | **9.6** | **20.1** | **8.8** | **20.8** |

### 4.5 DISCUSSION

CutOnce does not show an advantage over SOTA methods. However, after using the pseudo masks for training, our method demonstrates a certain advantage. As shown in Table 2, CutOnce produces the fewest masks but achieves an acceptable AP. Although VoteCut (Arica et al., 2024) leads across all metrics, its predictions *contain a large number of inaccurate annotations*, which can negatively impact model training.

Table 7: **Number of self-training rounds in COLER.**

| Round | COCO | | | | KITTI | |
|---|---|---|---|---|---|---|
| | $AP^{box}$ | $AP^{box}_{50}$ | $AP^{mask}$ | $AP^{mask}_{50}$ | $AP^{box}$ | $AP^{box}_{50}$ |
| 0 | 12.2 | 23.2 | 9.2 | 19.6 | 7.6 | 18.8 |
| 1 | **12.5** | **23.8** | 9.6 | **20.1** | 8.8 | **20.8** |
| 2 | 12.3 | 23.7 | **9.8** | 19.9 | **8.9** | 20.6 |

Despite the strong performance of COLER, it also has several limitations: *1)* Our approach heavily relies on the SSM, which is not fundamentally different from previous methods. The performance of COLER depends on the pseudo-labels, whose quality is determined by the pre-trained SSM. *2)* When multiple objects in an image are highly overlapping, COLER still struggles to separate them effectively. It also fails to handle occluded objects. *3)* In some cases, CutOnce can distinguish spatially adjacent objects, whereas COLER fails to do so. To address these limitations, we plan to *explore end-to-end unsupervised object discovery* in future work.

## 5 CONCLUSION

We propose CutOnce, a novel training-free method for unsupervised object discovery that efficiently and accurately partitions multiple instances within a single image. In particular, the *boundary augmentation strategy stands out as the simplest yet most effective improvement* in this work, and we believe it holds great potential for broader applications in the future. We also introduce COLER, a zero-shot model trained using masks generated by CutOnce. With only ImageNet-1K as the source domain and relatively low cost, COLER surpasses previous SOTA models on multiple benchmarks.

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

# A APPENDIX

## A.1 DATASETS USED FOR EVALUATION

**COCO** Lin et al. (2014) (Microsoft Common Objects in Context) is a large-scale dataset for object detection and segmentation. In this paper, COCO refers to the 5k images from the `2017 validation` set.

**COCO 20K** Lin et al. (2014) contains 19,817 images, a subset of COCO train2014. Many previous unsupervised methods Wang et al. (2022b; 2023); Arica et al. (2024) have used this dataset to evaluate model performance.

**LVIS** Gupta et al. (2019): (Large Vocabulary Instance Segmentation) is a dataset for long-tail instance segmentation. It contains 2.2 million high-quality instance masks of over 1,000 entry-level object categories, collected based on the COCO dataset. In this paper, LVIS refers to the 19,809 images in the `validation` set.

Table 8: **Summary of datasets used for zero-shot evaluation (except ImageNet).** "avg. # obj." denotes the average number of annotations per image.

| datasets | testing data | seg label | #images | avg. # obj. |
|---|---|---|---|---|
| COCO | val2017 | ✓ | 5,000 | 7.4 |
| COCO20K | train2014 | ✓ | 19,817 | 7.3 |
| LVIS | val | ✓ | 19,809 | 12.4 |
| Pascal VOC | trainval07 | ✗ | 9,963 | 3.1 |
| KITTI | trainval | ✗ | 7,521 | 4.7 |
| OpenImages V7 | val | ✗ | 41,620 | 7.3 |
| Object365 V2 | val | ✗ | 80,000 | 15.5 |
| ImageNet | val | ✗ | 50,000 | 1.6 |

**VOC** Everingham et al. (2010) (PASCAL Visual Object Classes) is a widely used benchmark for object detection. We evaluate on its `trainval07` split.

**KITTI** Geiger et al. (2012) (Karlsruhe Institute of Technology and Toyota Technological Institute) is one of the most popular datasets for mobile robotics and autonomous driving. We evaluate on its `trainval` split.

**OpenImages V7** Kuznetsova et al. (2020) contains multiple tasks, including image classification, object detection, instance segmentation, and visual relationship detection. We evaluate on over 40K images from the `val` split.

**Object365 V2** Shao et al. (2019) provides a supervised object detection benchmark with a focus on diverse objects in the natural world. We evaluate on 80K images from the `val` split.

The summary of these datasets used for zero-shot evaluation is provided in Table 8.

Table 9: **Unsupervised instance segmentation results on all benchmarks in this work.**

| Datasets | $AP^{mask}$ | $AP_{50}^{mask}$ | $AP_{75}^{mask}$ | $AP_S^{mask}$ | $AP_M^{mask}$ | $AP_L^{mask}$ | $AR_1^{mask}$ | $AR_{10}^{mask}$ | $AR_{100}^{mask}$ | $AR_S^{mask}$ | $AR_M^{mask}$ | $AR_L^{mask}$ |
|---|---|---|---|---|---|---|---|---|---|---|---|---|
| COCO | 9.6 | 20.1 | 8.5 | 2.3 | 10.5 | 21.5 | 5.6 | 16.3 | 25.4 | 9.5 | 31.2 | 44.9 |
| COCO20K | 9.8 | 20.5 | 8.4 | 2.6 | 10.5 | 21.6 | 5.6 | 16.5 | 25.6 | 9.7 | 31.6 | 44.7 |
| LVIS | 3.7 | 7.3 | 3.2 | 1.5 | 6.9 | 12.0 | 2.1 | 7.9 | 16.1 | 6.3 | 29.2 | 41.8 |

Table 10: **Unsupervised object detection results on all benchmarks in this work.**

| Datasets | $AP^{box}$ | $AP_{50}^{box}$ | $AP_{75}^{box}$ | $AP_S^{box}$ | $AP_M^{box}$ | $AP_L^{box}$ | $AR_1^{box}$ | $AR_{10}^{box}$ | $AR_{100}^{box}$ | $AR_S^{box}$ | $AR_M^{box}$ | $AR_L^{box}$ |
|---|---|---|---|---|---|---|---|---|---|---|---|---|
| COCO | 12.5 | 23.8 | 11.9 | 4.0 | 13.5 | 27.7 | 6.6 | 19.8 | 32.0 | 13.2 | 38.7 | 55.8 |
| COCO20K | 12.6 | 24.1 | 11.8 | 4.3 | 13.3 | 27.6 | 6.6 | 20.0 | 32.2 | 13.5 | 38.9 | 55.8 |
| LVIS | 4.6 | 9.2 | 4.1 | 2.4 | 8.6 | 15.5 | 2.4 | 9.5 | 20.0 | 8.7 | 34.9 | 51.6 |
| VOC | 20.5 | 39.1 | 19.6 | 2.7 | 8.2 | 31.5 | 15.9 | 33.2 | 44.6 | 19.3 | 36.5 | 54.5 |
| KITTI | 8.8 | 20.8 | 6.2 | 1.2 | 6.2 | 17.4 | 6.3 | 19.7 | 29.7 | 17.4 | 26.8 | 42.1 |
| OpenImages | 9.3 | 16.7 | 9.2 | 0.2 | 1.9 | 14.6 | 6.6 | 16.5 | 27.1 | 4.1 | 19.6 | 34.6 |
| Objects365 | 11.2 | 22.6 | 9.8 | 2.6 | 10.5 | 19.1 | 2.8 | 15.0 | 32.0 | 11.7 | 34.6 | 45.9 |

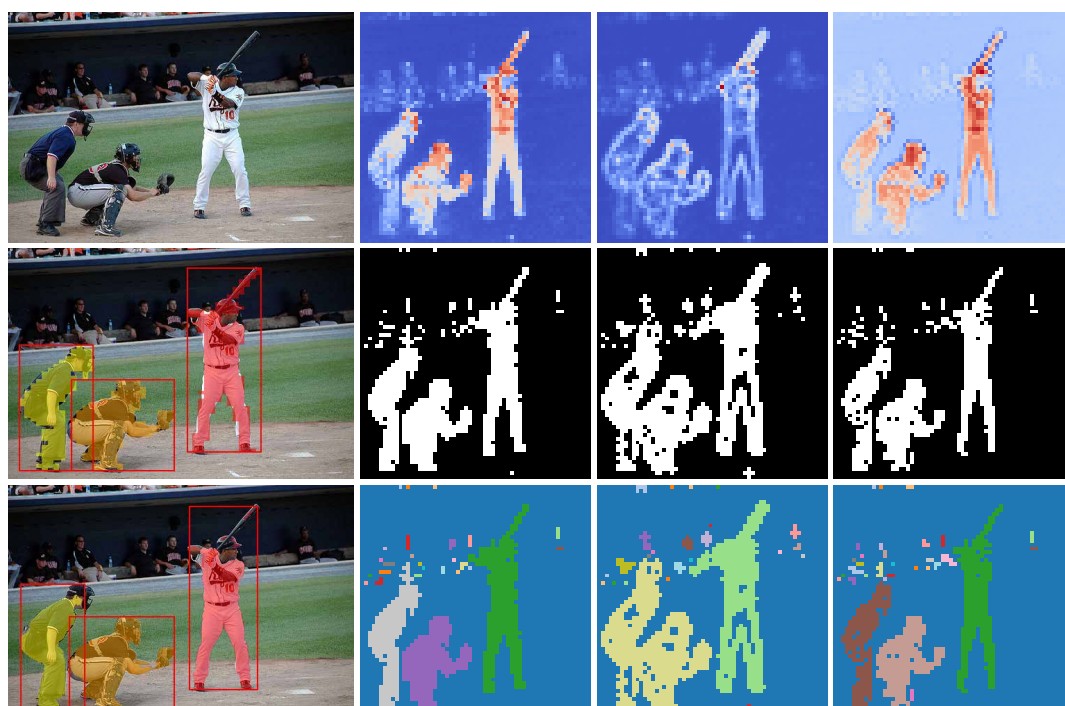

Figure 5: **Visualization of the computation process of CutOnce.** The first column (top to bottom) shows the original image, CutOnce prediction, and CutOnce with post-processing. The second to fourth columns show *raw eigenvector*, *boundary eigenvector*, *difference between the two*, and the corresponding foreground-background *binary maps* and *connected component maps*.

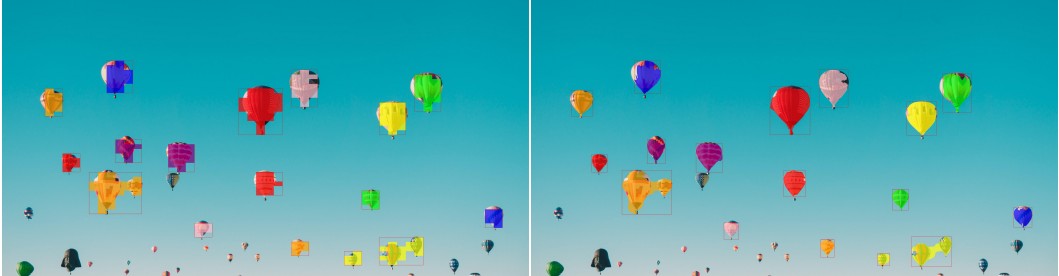

Figure 6: **Detecting many objects with CutOnce**. The first row shows the results of CutOnce, and the second row presents the results of CutOnce with post-processing. Both methods successfully detect *17 objects*.

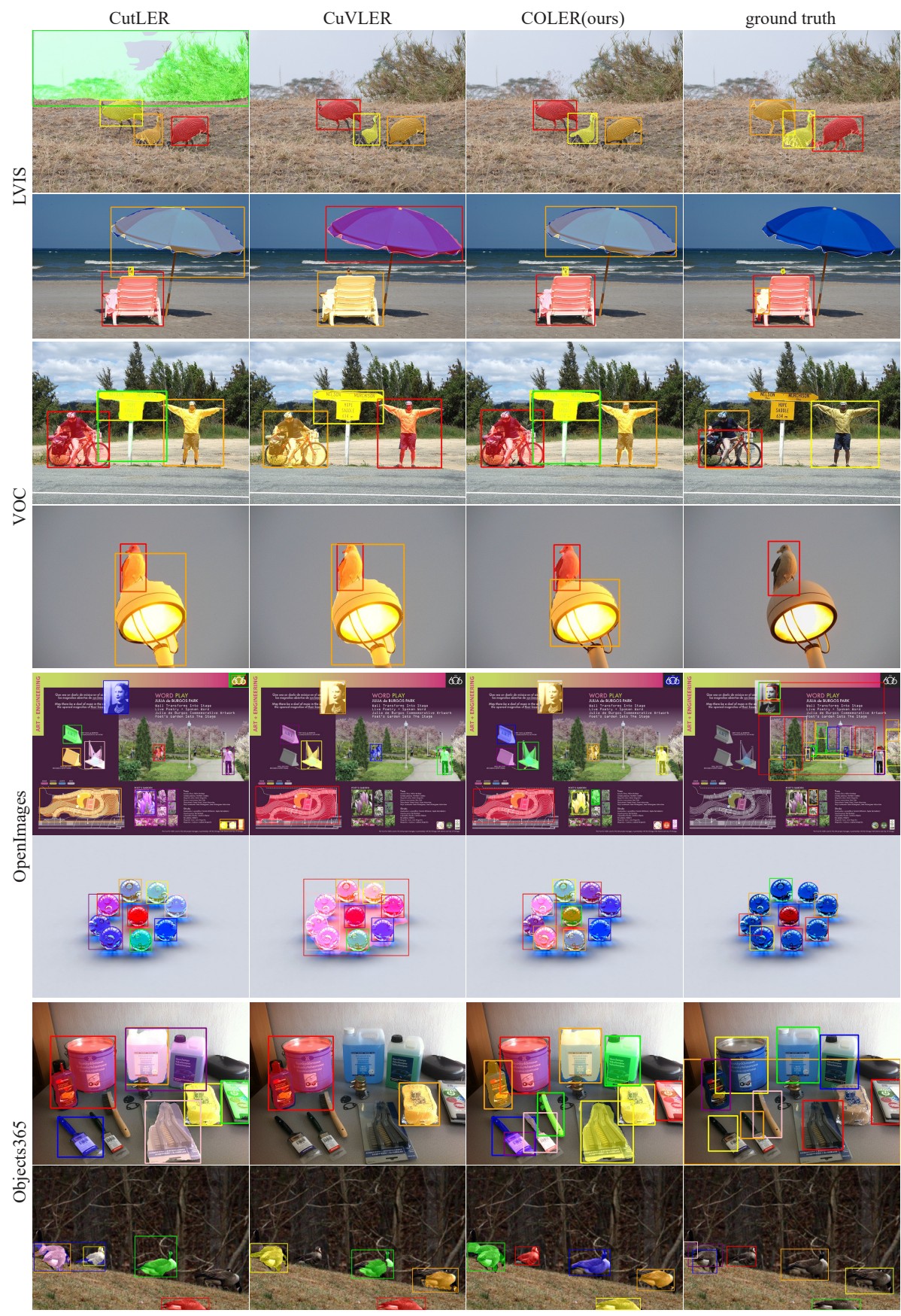

Figure 7: **Qualitative comparison of our COLER previous SOTA methods on LVIS, VOC, OpenImages, and Objects365.**

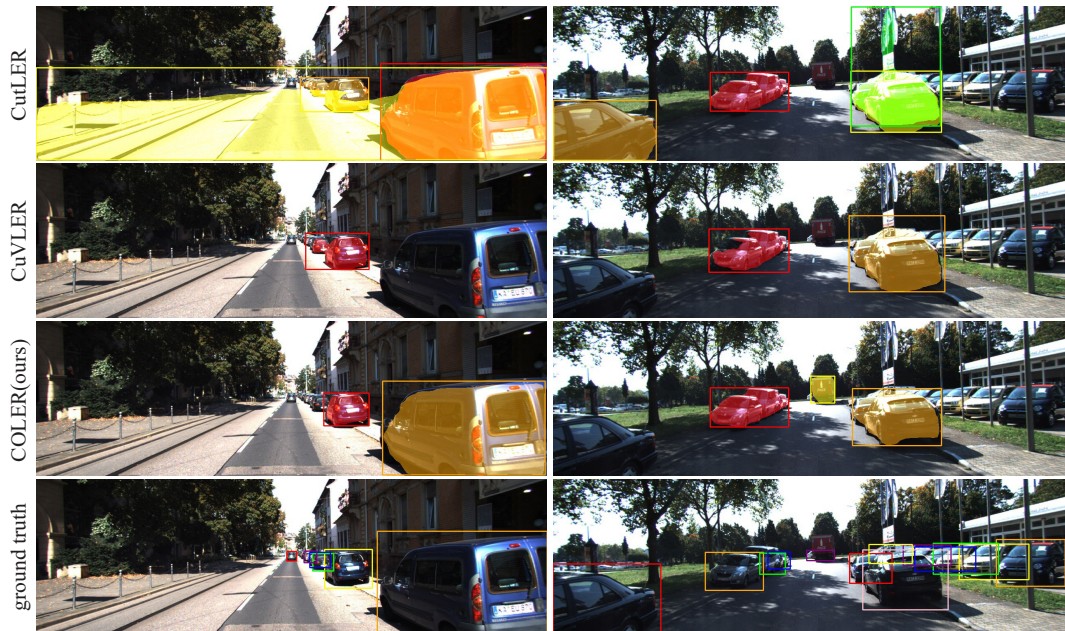

Figure 8: **Qualitative comparison of our COLER with previous SOTA methods on KITTI.**

## A.2 OTHER VISUALIZATIONS

Figure 5 shows a visualization of the intermediate computation process of *CutOnce's boundary enhancement module*. Obviously, the mechanism of this module is easy to understand and shows immediate effectiveness.

Figure 6 demonstrates the strong capability of CutOnce in segmenting multiple objects, which previous methods were unable to detect in such quantity.

Table 9 and Table 10 present the zero-shot evaluation results on unsupervised instance segmentation and object detection tasks across various datasets, respectively.

Figure 8 and Figure 7 show additional visualization results of our COLER method compared to previous state-of-the-art approaches. These figures only display predicted results with a confidence score of *no less than 0.5* (ground truth is excluded).

