# OpenReview forum: "Enhancing Object Discovery for Unsupervised Instance Segmentation and Object Detection"
_ICLR.cc/2026/Conference — Submitted to ICLR 2026_

### Official Review · Reviewer_HFo2 · 2025-10-24

**Soundness:** 3
**Presentation:** 3
**Contribution:** 2
**Rating:** 4
**Confidence:** 3

**Summary:**

Building on existing methods (as the authors themselves note) this paper presents a method for unsupervised object discovery. The method is based on normalized cuts of the graph Laplacian calculated using pre-trained image features (DINO, in this case). The main contributions of the proposed method is a boundary enhancing procedure based on the difference the second eigevector and a boundary vector obtained by calculating the distance between each feature and its neighbours. In practice this is a form of contrast enhancement (typical in image processing). The resulting boundary enhanced map provides cleaner and more detailed segments. Two other post-processing steps are proposed (rank based filtering which gets rids of noisy areas and self-training). Some variatns of the model use a CRF to further refine results. The main advantgage of the method is that it produces several object proposal while still needing to perform the Ncut once.
The mehod is evaluated on several image segmentation datasets and is shown to improve results.

**Strengths:**

The paper proposes some nice engineering solutions to problems faced by segmentation methods. Results are mildly better than proposed baselines and the method seems to be cheaper to run (as discussed in the paper).
The experiments are extensive enough with some ablation studies and a nice limitations section.

**Weaknesses:**

My main concern regarding this paper is its significance. Now, I am not an expert in the field, but to me this seems like a series of ad-hoc improvements to existing methods. I don't claim this is not important, or that the results are invalid - but I do feel that as a scientific paper I would expect to learn something new when reading. I do not feel that this paper, beyond the obvious technical contributions, have taught me anything.

Furthermore, even taking into account the scope of the paper I feel the experimental results, while fine, are underwhelming and improvements are minimal.

I am also wondering why the second round of self-training hurts performance on almost all benchmarks? can the authors elaborate?

One last remark - while the presentation of the paper is fine and it is readable, I found the use of SSM as abbreviation for a "self supervised model" extremely confusing (as SSM usually refers to state-space models).

**Questions:**

See above.

---

> ### Author Response · Authors · 2025-11-27
> **Author Rebuttal**
>
> We sincerely appreciate your careful review of our manuscript. We are pleased to address your questions and concerns, and will incorporate key points into the revised version of the paper.
>
> Revised manuscript link: https://anonymous.4open.science/r/CutOnce-68E3/iclr2026-rebuttal.pdf
>
> ## Weakness 1
> We optimize the application of NCut in multi-object segmentation, an approach not explored by other comparable methods. Our CutOnce achieves a computational complexity and running speed close to those of TokenCut. Our breakthrough contributions are summarized as follows:
> 1. We improved the NCut algorithm, **pioneering a new approach for multi-object segmentation (by optimizing the NCut spectra)**. The design of our first two modules aligns with the principles of several classical image processing algorithms, thereby enhancing the theoretical foundation. Specifically, Module 1 shares similar ideas with **self-tuning spectral clustering**[1], and Module 2 is analogous to the **Laplacian of Gaussian (LoG)[2] edge detection algorithm**.
> 2. To the best of our knowledge, **no existing methods** have adopted a similar framework—previous approaches rely on recursive NCut or clustering.
> 3. Our method can be **easily integrated into any methods that use NCut for processing self-supervised features** (e.g., MaskCut, VoteCut, DiffCut) to boost their final performance.
>
> In summary, we have successfully **extended the applicability of the traditional NCut algorithm beyond the original authors’ initial design**. Compared to methods that use NCut without any optimizations (e.g., CutLER, CuVLER, unMORE) which add effective but time-consuming modules after NCut, **our approach is clearly distinctive**.
>
> ## Weakness 2
> In the original manuscript, the performance improvement of COLER over CuVLER is greater than that of CuVLER over CutLER. In the revised version, we trained COLER on ImageNet Train, achieving stronger performance than the original version.
> Table 4 shows that our COLER **achieves 10% and 6% gains in $AP_{50}$ and AP on average**, respectively, compared to the SOTA, with improvements of up to 20% on certain datasets.
>
> ## Weakness 3
> We are not certain about the exact cause of this issue, but based on existing experiments, it is likely due to insufficient training data. ImageNet Val only contains 50K images, and our self-training setup uses a batch size of 8 with 60K iterations.
>
> When we trained on ImageNet Train (26 times the size of ImageNet Val), the performance became comparable to that of CutLER, and **no further improvement was observed after the third training round**.
>
> ## Weakness 4
>
> To avoid ambiguity, we accept the reviewer’s comments.
>
> ## References
> [1] Lihi Zelnik-Manor and Pietro Perona. Self-tuning spectral clustering. Advances in neural information processing systems, 17, 2004.
> [2] David Marr and Ellen Hildreth. Theory of edge detection. Proceedings of the Royal Society of London. Series B. Biological Sciences, 207(1167):187–217, 1980.

---

### Official Review · Reviewer_okBi · 2025-10-27

**Soundness:** 1
**Presentation:** 2
**Contribution:** 1
**Rating:** 2
**Confidence:** 5

**Summary:**

This paper focuses on the unsupervised instance segmentation task.
It first proposes the CutOnce module to generate pseudo labels. More specifically, CutOnce first performs N-Cut on pretrained self-supervised features, followed by boundary augmentation and ranking.
Then, the COLER module uses pseudo-labels generated from CutOnce to train the object detector.

**Strengths:**

The presentation of this paper is clear.
Writing logic is easy to follow, and illustrations are helpful for understanding.

**Weaknesses:**

1. The proposed method cannot fundamentally solve the issues mentioned in the paper (i.e. more accurate multi-object segmentation):
- As long as pseudo-labels are directly generated by N-Cut self-supervised features and the SSL features do not reveal real objectness, heuristic tricks like boundary processing, connected-component analysis can only receive incremental improvement, but not fundamentally solve the problem.

2. The designs of some modules are not well justified:
- First 2 modules in CutOnce are designed for "enhancing the distinction between foreground and background" as claimed in line 183.
- If the design only serves for foreground-background contrast, it is not enough for the goal to "enable NCut algorithm to discover multiple objects rather than focusing on a single one" as mentioned in line 179.

3. The contribution of this paper to the relevant field is minor:
- It follows the practice of CutLER and CuVELR, only introducing some tricks for processing self-supervised features.
- Compared with baselines, the performance gain is marginal.

4. Some important related works are not discussed and compared:
- unMORE: Unsupervised Multi-Object Segmentation via Center-Boundary Reasoning (icml25)

5. Some experiment settings are problematic:
- Pretraining datasets are not consistent for different methods (some on IN train, others on IN val).
- CuVLER are not given self-training.
- AR metrics are not calculated from the final detection evaluation.

**Questions:**

1. Fundamentally, why is Boundary Augmentation helpful for multi-object segmentation?
- Why "the saliency of more objects is enhanced"? From equation (5), what you are doing is more like an averaging operation within the neighborhood. For the visualization in Figure 1, can you normalize the heatmap before and after the boundary augmentation to facilitate a more direct comparison?
- Will it give a similar effect as the proposed module if one simply applies mask erosion onto the original binary mask in order to make  "Nearby objects are less likely to be considered as a single entity"?

2. Does the "boundary" mentioned in the Boundary Augmentation module refer to the boundary of objects? Or just the boundary of the foreground? If two objects are tightly adjacent to each other (i.e., there are no background pixels between them), can they be segmented into two?

3. For "Ranking-Based Instance Filter", why use feature sum as the sorting criterion? What does the feature sum value represent?

4. In Table 3, why are CuVLER and COLER pretrained on the ImageNet validation set instead of the training set? It is important to align all settings to make the experimental results comparable.

5. In Table 3, why not provide CuVLER with self-training while all other methods are doing so? In addition, the original CuVLER paper also uses self-training and yields better AP50 (23.5) than you provide in Table 3 (23.0).

6. Table 3 should also include AR.

7. It is claimed that COLER may detect instances that are not annotated in Ground Truth, consider evaluating on COCO* mentioned in unMORE (https://arxiv.org/abs/2506.01778).

8. Given that pseudo-labels generated from CutOnce are obviously worse than VoteCut as suggested in Table 2, why do detectors trained on CutOnce yield better performance than those trained on VoteCut as suggested in Table 3?

---

> ### Author Response · Authors · 2025-11-27
> **Author Rebuttal**
>
> Given that you have raised some questions lacking in-depth consideration, we suggest that you carefully review the relevant literature in this field before evaluating our manuscript.
>
> **If you are unable to understand the domain-specific knowledge presented in the paper, please clarify to all parties that you are not familiar with this research area and lower your confidence score accordingly.**
>
> Revised manuscript link: https://anonymous.4open.science/r/CutOnce-68E3/iclr2026-rebuttal.pdf
>
> Our experiments were conducted on a single NVIDIA 4090 24G GPU, with a severely constrained computing power configuration compared to similar methods (at least five times that of ours). In the original version, we trained on ImageNet Val. In the revised version, we use ImageNet Train (26 times the number of images in Val) for training, while the ablation experiments still adopt the CutOnce version trained on ImageNet Val.
>
> ## Weakness 1
> We optimize the application of NCut in multi-object segmentation, an approach not explored by other comparable methods.
> Our CutOnce achieves a computational complexity and running speed close to those of TokenCut. Our breakthrough contributions are summarized as follows:
> 1. We improved the NCut algorithm, **pioneering a new paradigm for multi-object segmentation (by optimizing the NCut spectra)**. The design of our first two modules aligns with the principles of several classical image processing algorithms, thereby enhancing the theoretical foundation. Specifically, Module 1 shares similar ideas with **self-tuning spectral clustering**[1], and Module 2 is analogous to the **Laplacian of Gaussian (LoG)[2] edge detection algorithm**.
> 2. To the best of our knowledge, no existing methods have adopted a similar framework—previous approaches rely on recursive NCut or clustering.
> 3. Our method can be **easily integrated into any methods that use NCut for processing self-supervised features** (e.g., MaskCut, VoteCut, DiffCut) to boost their final performance.
>
> In summary, we have successfully **extended the applicability of the traditional NCut algorithm beyond the original authors’ initial design**.
>
> ## Weakness 2
> We first briefly summarize the working mechanism of the first two modules in CutOnce:
>
> - **Density-tune Module**: A smoother affinity matrix ( W ) (affinity dispersed among more nodes) → more stable eigenvector → clearer segmentation boundaries.
> - **Boundary Augmentation Module**:
>   1. Enhances the contrast between boundaries and targets (i.e., foreground-background contrast) → sharper gradient of the eigenvector at real boundaries → improved separability of multiple objects.
>   2. Reduces internal differences within targets (appropriately minimizing internal target features) → moderate expansion of foreground regions → increased focus on more targets.
>
> In summary, both modules enhance the distinguishability between foreground and background. Boundary augmentation enables NCut to focus on multiple targets (Figures 1, 2, 5) and prevents closely spaced objects from being misidentified as a single target.
>
> ## Weakness 3
> Please refer to Weakness 1 for our contributions.
>
> Due to GPU computing power constraints, the performance improvement in our original submission was not sufficiently significant. In the revised version, we trained COLER on ImageNet Train, achieving stronger performance than the original version.
> Table 4 shows that our COLER **achieves 10% and 6% gains in $AP_{50}$ and AP on average**, respectively, compared to the SOTA, with improvements of up to 20% on certain datasets.
>
> ## Weakness 4
> unMORE (ICML 2025) is not included for comparison because **it does not support zero-shot evaluation**. It uses two datasets (ImageNet and COCO) for training instead of only ImageNet, which is clearly unfair.
>
> ImageNet does not contain manually annotated segmentation labels (the Val split only includes box labels), while COCO provides such labels. **Thus, CutLER, CuVLER, and our COLER are more appropriately regarded as true unsupervised methods**.
>
> ## Weakness 5
> We used ImageNet Val to accelerate our experiment progress. In the revised version, we found that **training COLER on ImageNet train yields significantly better results**.
>
> **CuVLER’s official implementation also uses ImageNet Val**, which allows testing the performance metrics of pseudo-masks.
>
> CuVLER’s official self-training uses the COCO dataset, which **violates the zero-shot evaluation protocol**. *We attempted self-training using their source code but found that the performance was inferior to CutLER*.
>
> The AR evaluation for pseudo-labels uses COCO’s official API, and the AR evaluation for the model part adopts the Detectron2 framework. **Please clarify any aspects where our approach is deemed incorrect—we have provided the source code in the supplementary materials.**
>
> ## Continued rebuttal due to word limit for comments ……

---

> ### Author Response · Authors · 2025-11-27
> **Author Rebuttal (Continued with the previous comment)**
>
> ## Question 1
> *The direct manifestation of segmentation quality is the accurate detection of boundaries.* This is precisely the role of the Boundary Enhancement Module, which is similar to the classical image processing algorithm LoG. **Both capture local intensity changes** (the former is an approximation of a first-order gradient of the NCut eigenvector, while the latter is the sum of second-order derivatives) **for edge detection**.
>
> The suggestion to plot heatmaps using normalized vectors is valuable, **and we have adopted it**. However, we found that all heatmaps before and after normalization are almost identical, indicating that the current visual verification (i.e., boundary enhancement significantly improves segmentation) is valid.
>
> In fact, beyond heatmaps, the effect of boundary enhancement can be better understood through the corresponding binary maps and connected component maps of the heatmaps, **which are provided in the appendix of the original paper**.
>
> Erosion operations only shrink mask regions spatially and are **fixed morphological operations** that **cannot be adaptively adjusted based on feature, density, or boundary information**. Thus, they cannot achieve the effect of our module. Figure 4 demonstrates the good adaptability of our two enhanced object discovery modules. Our method dynamically models the separability between objects in the feature space, rather than performing static morphological modifications on pixels—hence, the two approaches differ in working mechanism and effectiveness.
>
> ## Question 2
> Boundaries refer to the **boundaries of all targets**, i.e., the **boundaries of the foreground.**
>
> If two objects are closely adjacent, they are likely to be missegmented as a single object. However, if the self-supervised model can detect the "boundary" between these two objects (characterized by large feature differences), our module will enhance the contrast between this boundary and the targets, enabling successful segmentation. *The self-supervised model **determines the upper limit of detection capability**; TokenCut introduces NCut to bring the detection capability **closer to this upper limit**, and our optimization of NCut **further narrows the gap**.*
>
> ## Question 3
> Previous methods (TokenCut, CutLER) determine a target **based on the feature magnitude of a single point**, which is prone to misjudgment (as noted in the original TokenCut paper). Specifically, they select the connected component containing the maximum absolute eigenvector value as the only target. In contrast, we **use the cumulative feature magnitude of a region instead of a single point to determine** which target is prioritized by NCut. This approach **considers both the spatial coverage of the target and its degree of attention** when selecting salient targets.
>
> Each coordinate on the eigenvector corresponds to a patch extracted by ViT, and each patch has a feature value computed by NCut. Each target is a connected component corresponding to multiple adjacent patches. *The total feature value refers to the sum of all feature values of the patches in the region where each target is located*.
>
> ## Question 4
> We used ImageNet Val to accelerate our experiment progress, the ratio of the number of images in ImageNet Train to Val is 128:5. CuVLER’s official implementation also uses ImageNet Val, which allows testing the performance metrics of pseudo-masks. When we trained CuVLER using either ImageNet Train or Val with the official code, we were unable to replicate the performance of CuVLER’s pre-trained weights. In the revised version, we found that **training COLER on ImageNet train yields significantly better results**.
>
> ## Question 5
> CuVLER’s official self-training uses the test dataset (COCO) instead of the training dataset (ImageNet), thus **failing to meet the requirements of zero-shot evaluation**. For fair comparison, we have **made every effort to present the best performance of other methods**. As mentioned in Question 4, we were unable to achieve better performance for CuVLER.
>
> The note *below Table 3 in CuVLER’s original paper indicates that self-training was performed on each test dataset*. In contrast, CutLER and our COLER were **self-trained on ImageNet** before being tested on all datasets. *We report a value of 23.0 instead of 23.5 because we used the zero-shot weights provided by CuVLER’s official implementation*.
>
> ## Question 6
> CutLER and CuVLER do not use the AR metric, with AP being the primary focus. For part-level segmentation (e.g., UnSAM), AR should be the main metric of concern.
>
> ## Question 7
> The COCO* annotation set has not been widely recognized, so we do not consider using it. COLER can detect instances not annotated in the Ground Truth, as evidenced by Figure 3.
>
> ## Continued rebuttal due to word limit for comments ……

---

> ### Author Response · Authors · 2025-11-27
> **Author Rebuttal (Continued with the previous comment)**
>
> ## Question 8
> Before addressing this question, we pose the following: *Why does MaskCut perform much worse than VoteCut in terms of intermediate results, yet their final performance is quite similar?*
>
> We have provided a response in Section 4.5 DISCUSSION of the original paper. As shown in Table 2, VoteCut segments 9 labels per image, far more than other methods, but the improvements in AP and AR are very limited. **Although AP is penalized by incorrect masks, a large number of false labels are still introduced**.
>
> In the revised paper, we display all predicted masks in the VoteCut row of Figure 2 instead of only those with a confidence score greater than 50%. It is evident that **VoteCut introduces numerous false labels**, requiring special loss functions for training (CutLER’s DropLoss and their own soft target loss), whereas our CutOnce does not require any specific loss functions.
>
> ## References
> [1] Lihi Zelnik-Manor and Pietro Perona. Self-tuning spectral clustering. Advances in neural information processing systems, 17, 2004.
> [2] David Marr and Ellen Hildreth. Theory of edge detection. Proceedings of the Royal Society of London. Series B. Biological Sciences, 207(1167):187–217, 1980.

---

### Official Review · Reviewer_efiD · 2025-10-30

**Soundness:** 3
**Presentation:** 3
**Contribution:** 2
**Rating:** 6
**Confidence:** 4

**Summary:**

This paper proposes COLER (Cut-Once-and-LEaRn) for unsupervised, zero-shot instance segmentation and object detection. The pipeline has two stages: 1. CutOnce (train-free pseudo-mask generator). It applies Normalized Cut (NCut) only once on DINO-ViT features, then adds three simple modules to (a) adapt similarity with density-tuned temperature (b) sharpen and separate instances via a boundary-augmentation transform on the NCut eigenvector, and (c) select foreground components using a ranked connected-component filter. Unlike TokenCut/MaskCut/VoteCut families that use recursive cuts or clustering, CutOnce claims multi-instance discovery from a single NCut pass. 2. LEaRn (detector training): it trains a class-agnostic Cascade Mask R-CNN solely on CutOnce masks and improves quality via one self-training round, deliberately avoiding bespoke pseudo-label losses.

**Strengths:**

- The method is technically sound and implementation-oriented: the density-tuned similarity (adaptive temperature), boundary-augmented eigenvector, and rank-based component selection are specified mathematically and ablated individually and cumulatively.

- The pipeline figure and intermediate visualizations (raw/“boundary”/difference eigenvectors, component maps) explain why single-pass NCut can still separate multiple instances. And, the writing is direct; notation for the three modules is compact and consistent.

- Single-pass NCut → multi-instance with three lightweight modules is a clear conceptual departure from recursive partition (MaskCut/CutLER) and clustering/voting (VoteCut/CuVLER). The design removes the need to guess cluster counts and avoids recursive error accumulation.

- Comparisons are made to strong open baselines: CutLER (MaskCut+detector+self-train) and CuVLER (VoteCut+detector+soft loss), showing COLER is competitive while being simpler and faster at the pseudo-mask stage.

**Weaknesses:**

- The three modules—adaptive temperature on cosine affinities, boundary emphasizing via neighborhood differencing, and rank filtering—are spectral pre/post-processing heuristics layered on a classical NCut pipeline (no new learning principle or theory). In contrast, DiffCut offers a more substantive change of backbone (diffusion UNet features) *and* a recursive NCut with granularity control; DiffNCut explores differentiability for end-to-end learning. The paper’s “SOTA” claim should be carefully bounded by setting/task

- Even with boundary augmentation, one eigenvector is a coarse partition; multi-object scenes with strong co-occurrence/overlap can remain merged. This is exactly what recursive strategies (MaskCut/CutLER; DiffCut) were built to address. The paper acknowledges multi-object limits; quantitative stress-tests on dense scenes (LVIS/Objects365) against recursive/voting approaches would sharpen the picture.

- The paper compares with CutLER/CuVLER, but omits direct head-to-head on DiffCut under an instance-level protocol (even a proxy comparison would help). Also, some CuVLER strengths (multi-SSL diversity, mask scoring) are not ablated against COLER’s single-backbone design.

- There is an appealing spectral intuition, but no theoretical insight into why the boundary augmentation (subtract neighborhood mean absolute difference) and density temperature scaling improve the eigenvector’s separability or component stability.

**Questions:**

- Why one eigenvector? Provide analysis on when the second smallest eigenvector is sufficient for multi-instance separation. An experiment using multi-eigenvector embeddings (top-k eigvecs with mean-shift or connected components) as a control could clarify the tradeoff versus your boundary augmentation.

- How about high-res inputs? Could you please show scaling experiments (e.g., 640², 1024²)? Does single-pass NCut remain stable and fast?

---

> ### Author Response · Authors · 2025-11-27
> **Author Rebuttal**
>
> We sincerely appreciate your careful review of our manuscript. We are pleased to address your questions and concerns, and will incorporate key points into the revised version of the paper.
>
> Revised manuscript link: https://anonymous.4open.science/r/CutOnce-68E3/iclr2026-rebuttal.pdf
>
> ## Weakness 1
> We acknowledge that the three modules are improvements based on NCut. Our CutOnce achieves a computational complexity and running speed close to those of TokenCut. Our breakthrough contributions are summarized as follows:
> 1. We improved the NCut algorithm, **pioneering a new approach for multi-object segmentation (by optimizing the NCut spectra)**. The design of our first two modules aligns with the principles of several classical image processing algorithms, thereby enhancing the theoretical foundation. Specifically, Module 1 shares similar ideas with **self-tuning spectral clustering**[1], and Module 2 is analogous to the **Laplacian of Gaussian (LoG)[2] edge detection algorithm**.
> 2. To the best of our knowledge, **no existing methods** have adopted a similar framework—previous approaches rely on recursive NCut or clustering.
> 3. Our method can be **easily integrated into any methods that use NCut for processing self-supervised features** (e.g., MaskCut, VoteCut, DiffCut) to boost their final performance.
>
> In summary, we have successfully **extended the applicability of the traditional NCut algorithm beyond the original authors’ initial design**. Compared to methods that use NCut without any optimizations (e.g., CutLER, CuVLER, unMORE) which add effective but time-consuming modules after NCut, **our approach is clearly distinctive**.
>
> DiffCut introduces a large-scale diffusion model as a feature extractor and uses clustering combined with recursive NCut. Its performance improvement stems from the **replacement of the feature extractor** (as compared in the paper), with limited innovations in the workflow compared to MaskCut. DiffNCut fine-tunes DINO on specific data and is designed for single-object rather than multi-object segmentation. It achieves only marginal improvements over training-free TokenCut (2022 CVPR) and **exhibits poor generalization**.
>
> We will adopt your suggestion and more rigorously define the SOTA claim in the paper as “the best Efficiency-Performance trade-off in zero-shot unsupervised instance segmentation tasks”.
>
> ## Weakness 2
> Compared to previous methods, even with rough segmentation, our boundary enhancement enables stronger distinguishability between closely spaced objects.
>
> Recursive strategies cannot solve the overlapping problem. **Existing recursive methods use the original NCut, which is less accurate in boundary recognition than the optimized NCut we propose**. The upper limit of the ability to distinguish overlapping objects depends on the self-supervised model, while the lower limit relies on the capability to mine self-supervised features. By using an NCut with enhanced boundary detection capability, CutOnce is more conducive to approaching the theoretical upper limit compared to MaskCut. Furthermore, the segmentation error of NCut increases with the number of recursive iterations. **This reasoning is verified by the first image in the MaskCut row of Figure 2**.
>
> Our COLER demonstrates advantages in dense scenarios, such as LVIS (Table 3) and Objects365 (Table 4), and also performs well in complex scenarios like KITTI. **Visual comparisons on these datasets are included in the supplementary materials.**
>
> ## Weakness 3
> The core task of DiffCut is unsupervised semantic segmentation, whose output does **not distinguish instance IDs** (multiple objects of the same category are merged). Therefore, direct comparison on the instance segmentation task is **infeasible**.
>
> ## Weakness 4
> The theory of Density-Tune can be linked to the stability of Spectral Clustering. In standard NCut, the interior of target regions often leads to extreme values of feature vectors in these areas (trivial cuts). This is similar to the idea in Self-tuning spectral clustering (2004 NeurIPS), where **the affinity matrix is optimized through neighborhood information to reduce sensitivity**.
>
> The direct manifestation of segmentation quality is the accurate detection of boundaries. Our $X_b$ is a **local difference (an approximation of a first-order gradient) computed from the eigenvector**, capturing *boundary-related variation patterns*. The idea is similar to the classical Laplacian of Gaussian (LoG) (Marr & Hildreth, 1980) in image processing, where strong local changes (the sum of second-order derivatives) *are used for edge detection*.
>
> In summary, both of our modules are highly consistent with classical algorithms in terms of core ideas.
>
> ## Continued rebuttal due to word limit for comments ……

---

> ### Author Response · Authors · 2025-11-27
> **Author Rebuttal (Continued with the previous comment)**
>
> ## Question 1
> The original NCut paper uses the second smallest eigenvector, and this setting is also adopted by subsequent works using NCut (e.g., CutLER, CuVLER). We conducted comparative experiments with other eigenvectors and found that the second smallest eigenvector yields the best performance. The table below reports the results of CutOnce on ImageNet Val.
>
> | eigenvector | $\mathrm{AP}_{50}$ | AP   | AR    |
> |-------------|---------|------|-------|
> | 1st         | 19.2    | 12.6 | 17.5  |
> | 2nd         | 32.5    | 16.5 | 31.5  |
> | 3rd         | 23.5    | 13.5 | 23.1  |
>
> The second smallest eigenvector can be directly used for multi-instance segmentation **without any additional constraints**.
>
> ## Question 2
>
> We follow the resolution setting of CutLER, which is also adopted by CuVLER. The results of CutOnce with different resolutions on COCOval 2017 are shown in the table below: while the segmentation accuracy of pseudo-masks is slightly improved, the image generation speed decreases significantly. The increased latency mainly comes from feature extraction by the self-supervised model and feature processing by NCut, with the latter contributing more to the time overhead. Since our experiments only use ImageNet data, which contains very few high-resolution images, we ultimately retain the size setting of 480.
>
> | Size        | 480  | 640  | 1024  |
> |-------------|------|------|-------|
> | $\mathrm{AP}_{50}$       | 19.6 | 19.8 | 19.9  |
> | Time(s/img) | 0.24 | 0.8  | 9     |
>
> ## References
> [1] Lihi Zelnik-Manor and Pietro Perona. Self-tuning spectral clustering. Advances in neural information processing systems, 17, 2004.
> [2] David Marr and Ellen Hildreth. Theory of edge detection. Proceedings of the Royal Society of London. Series B. Biological Sciences, 207(1167):187–217, 1980.

---

### Official Review · Reviewer_dVFE · 2025-11-01

**Soundness:** 2
**Presentation:** 2
**Contribution:** 2
**Rating:** 2
**Confidence:** 4

**Summary:**

This paper focuses on unsupervised instance segmentation and object detection. The authors introduce a pipeline that first generates pseudo labels using a self-supervised model, which are then used to train a standard detector. They propose several novel techniques during the pseudo label generation phase to ensure higher quality, validating the pipeline's strong performance across various datasets.

**Strengths:**

1. The presentation is clear and logical, effectively demonstrating the overall pipeline design with well-structured experiments.

2. The experimental results demonstrate strong performance, achieving favorable metrics when compared directly against the established baseline methods, suggesting the efficacy and competitive advantage of the proposed approach.

**Weaknesses:**

1. The overall pipeline lacks significant novelty, bearing a strong resemblance to existing unsupervised methods. Although the authors introduce several techniques to enhance pseudo label quality, the specific mechanisms—such as the methods used to refine the pseudo segmentation map and the filtering strategies—do not demonstrate a substantial departure from classical computer vision approaches.

2. The performance improvements achieved by the proposed method are marginal when compared to the existing state-of-the-art or baselines.

3. As the authors suggest, the method appears to heavily rely on the underlying self-supervised model. The reported gains achieved by simply switching or upgrading the self-supervised model are notably larger than the improvements resulting from the carefully designed pseudo-label strategies.

**Questions:**

See **Weakness**.

---

> ### Author Response · Authors · 2025-11-27
> **Author Rebuttal**
>
> We suggest that you carefully read the relevant literature in this field before reviewing our manuscript.
>
> **If you are unable to understand the domain-specific knowledge presented in the paper, please clarify to all parties that you are not familiar with this research area and lower your confidence score accordingly.**
>
> Revised manuscript link: https://anonymous.4open.science/r/CutOnce-68E3/iclr2026-rebuttal.pdf
>
> ## Weakness 1
> CuVLER (CVPR 2024), CutS3D (CVPR 2025), and our COLER share the core innovation of leveraging self-supervised features for pseudo-label generation. However, only COLER achieves **a substantial speedup in pseudo-label generation (approaching that of TokenCut)**. Our breakthrough contributions are summarized as follows:
> 1. We improved the NCut algorithm, **pioneering a new approach for multi-object segmentation (by optimizing the NCut spectra)**. The design of our first two modules **aligns with the principles of several classical image processing algorithms**, thereby enhancing the theoretical foundation.
> 2. To the best of our knowledge, **no existing methods** have adopted a similar framework—previous approaches rely on recursive NCut or clustering.
> 3. Our method can be **easily integrated into existing methods** (e.g., MaskCut, VoteCut) to boost their final performance.
>
> In summary, we have successfully extended the applicability of the traditional NCut algorithm **beyond the original authors’ initial design**.
>
> ## Weakness 2
> In the original manuscript, the performance improvement of COLER over CuVLER is greater than that of CuVLER over CutLER. In the revised version, we trained COLER on ImageNet Train, achieving stronger performance than the original version.
> Table 4 shows that our COLER **achieves 10% and 6% gains in $AP_{50}$ and AP on average**, respectively, compared to the SOTA, with improvements of up to 20% on certain datasets.
>
> In the updated manuscript, we trained our model on ImageNet-1K Train without using CRF post-processing, achieving higher performance than the original results (which used ImageNet-1K Val training and CRF post-processing). The limited performance in the original submission is explained as follows:
>
> Our experiments were conducted on a **single NVIDIA RTX 4090 GPU**, while other comparable methods used multiple A100 GPUs (with a computing power at least five times higher than ours). To accelerate the experiment cycle, we optimized the process in two aspects:
> - First, we **removed the CRF post-processing** step during pseudo-label generation. Unlike previous methods, our CutOnce module eliminates the dependency on CRF and still achieves high-quality boundary segmentation—this further validates the effectiveness of our two enhanced modules.
> - Second, we **used the ImageNet-1K Val split** (50K images) instead of the Train split (1.28M images) for training. Despite using fewer training data and no CRF post-processing, we still achieved competitive results. Clearly, additional training data and post-processing of pseudo-masks would further improve performance.
>
> CuVLER’s official implementation uses ImageNet-1K Val for training, and the results in our Tables 3 and 4 are based on their official zero-shot weights (the other set of weights is trained on COCO, which violates the zero-shot evaluation protocol). Thus, **our reported metrics for CuVLER are lower than those in the original paper**. Furthermore, since CuVLER does not provide weights trained on ImageNet-1K Train, we modified their source code and re-ran the experiments, but found that the results were inferior to the official zero-shot weights.
>
> ## Weakness 3
> Similar to existing methods (CutLER, CuVLER), COLER adopts self-supervised models **but innovates by optimizing the NCut algorithm**. Our CutOnce module achieves **a computational complexity close to TokenCut** while supporting multi-object segmentation.
>
> Methods similar to ours share this characteristic: they generate pseudo-labels using features from self-supervised models. Beyond the methods discussed in the paper, this also includes approaches such as UnSAM (NeurIPS 2024, fine-grained unsupervised instance segmentation) and CUPS (CVPR 2025, unsupervised panoptic segmentation).
>
> While switching to more advanced self-supervised models may yield greater gains, our experimental results demonstrate the value of optimizing the NCut algorithm: CutOnce **runs 10 times faster than state-of-the-art methods** and achieves significantly higher localization accuracy compared to counterparts.

---

### Author Response · Authors · 2025-11-26
**Revision Summary**

Dear Reviewers,

We have revised the manuscript based on your comments. The updated version mainly addresses two aspects:

Clarifying the novelty of our method.
We introduce improvements to the NCut algorithm and propose a new direction for multi-object segmentation by optimizing the NCut spectra. Our first two modules follow principles similar to classic image processing methods, which strengthens the theoretical foundation.
To the best of our knowledge, no existing work adopts a similar idea; prior methods typically rely on recursive NCut or clustering.
Our approach can be easily integrated into previous methods such as MaskCut and VoteCut, and it consistently improves their final performance.

Further improving performance.
By default, we use ImageNet train instead of ImageNet val for training (the former is 25 times larger). Models trained with ImageNet val are denoted as CutOnce* and COLER*.
Due to severe hardware limitations, the ablation experiments are still conducted using COLER*.

Revised manuscript:
https://anonymous.4open.science/r/CutOnce-68E3/iclr2026-rebuttal.pdf

We will respond to each reviewer’s questions as soon as possible and promptly update the link to the supplementary materials.

---

### Meta-Review · Area_Chair_VmyH · 2026-01-11

**Summary:**

COLER is proposed to improve self-supervised/unsupervised segmentation and detection by pseudo-labeling via normalized cuts. Experiments show accurate zero-shot performance on standard benchmarks for unsupervised object detection and instance segmentation incl. COCO, PASCAL VOC, Kitti, and others. Furthermore COLER only takes ~1/10th the time of state-of-the-art methods.

Four expert reviewers are split between acceptance (efiD: 6) and rejection (HFo2: 4, dVFE: 2, okBi: 2). The votes for rejection vary in their details but all share concerns about the novelty and significance for machine learning. While there is improvement, it may be too narrow in scope and application, and so it could not be informative enough a machine learning audience vs. a computer vision audience. The concerns are detailed further below. The vote for acceptance focuses on the technical soundness, computational efficiency, and  competitive results with strong and open baselines. It nevertheless shares similar concerns about the novelty and significance of the proposed modules to normalized cut. The meta-reviewer sides with rejection given the repeated concerns about novelty and significance that are not resolved by the rebuttal: the reiteration of the changes to normalized cut and the assertion that the method can be combined with others does not fundamentally change the scope of the technical contribution nor the included comparisons.

The authors are encouraged to incorporate the feedback from review and submit to a computer vision venue such as ECCV. This encouragement is derived from multiple reviewers not considering the technical improvement of the proposed method and its use of normalized cuts as a machine learning innovation of sufficient significance and generality for ICLR.

**Reviewer Concerns:**

- Novelty (dVFE, efiD, okBi): COLER follows CutLER CuVLER and others in its approach of pseudo-labeling based on a self-supervised representation. This work introduces techniques to improve pseudo-label quality, and do without some common and time-consuming post-processing like the dense CRF, but these are derived from established computer vision techniques. On the other hand the method of DiffCut (NeurIPS 2024) is more of a departure. As such, the novelty and technical contribution to machine learning is insufficient. The rebuttal argues that the optimization of the normalized cut spectra is novel and different from existing methods, and is the reason for COLER's efficiency.
- Significance (dVFE, efiD, okBi): The performance improvement is minor w.r.t. the state-of-the-art and the difference due to switching the input representation is larger than contributed by the proposed of unsupervised method on top of it. The rebuttal provides new results that make results more accurate, but less comparable, by changing the dataset for training. The rebuttal also further qualifies the SOTA claim to be about the trade-off in efficiency and performance to better respect existing results.
- Processing high-resolution inputs (efiD): Can COLER process high-resolution inputs with its single-pass Normalized Cut? That is, without recursive or multi-scale application as in other methods? The rebuttal provides results that achieve similar accuracy up to 1024^2 size but at >30x the time.

**Reviewer Scores:**

- dVFE: Likely to maintain score of 2 because the rebuttal mostly reiterates the points of the paper. The dismissive preamble of the rebuttal is unlikely conducive to a change in evaluation.
- okBi: Likely to maintain score of 2 or perhaps raise it to 3 because some points are clarified (like which potential comparisons would be in the same setting or not) and each question is given a reply. The dismissive preamble of the rebuttal is unlikely conducive to a change in evaluation.
- HFo2: Likely to maintain score of 4, but could flip to 6, because some questions like the effect of self-training are specifically answered. The points about what is learned and the significance of the results are addressed by the rebuttal in the same way for all reviewers, and may or may not be convincing since the response is a more detailed inventory of the technical specifics rather than a research question or statement of what has been learned (as requested by the reviewer).

---

### Decision · Program_Chairs · 2026-01-26

Reject